Recent research progress on the correlation between metabolic syndrome and Helicobacter pylori infection

Xie Qinli 1
He Yangjun 2
Zhou Danni 3
Jiang Yi 4
Deng Ying 5 1137432251@qq.com
Li Ruoqing 4 lrqlyx@foxmail.com
1 Department of Physical Examination Center, Chongqing University Central Hospital, Chongqing Emergency Medical Center, Chongqing Key Laboratory of Emergency Medicine , Chongqing , China
2 Department of Emergency, Chongqing University Central Hospital, Chongqing Emergency Medical Center, Chongqing Key Laboratory of Emergency Medicine , Chongqing , China
3 Center for Reproductive Medicine, Chongqing Health Center for Women and Children, Women and Children’s Hospital of Chongqing Medical University , Chongqing , China
4 Department of General Medicine, Chongqing University Central Hospital, Chongqing Emergency Medical Center, Chongqing Key Laboratory of Emergency Medicine , Chongqing , China
5 Department of Plastic Surgery, Chongqing University Central Hospital, Chongqing Emergency Medical Center, Chongqing Key Laboratory of Emergency Medicine , Chongqing , China
Qin Jiangjiang
Electronic publication date: 2023 Jul 19
Publication date: 2023
Volume: 11
Electronic Location ID: e15755
Received 2023 May 4; Accepted 2023 Jun 26
Copyright: © 2023 Xie et al.
Copyright year: 2023
Copyright holder: Xie et al.
License: This is an open access article distributed under the terms of the Creative Commons Attribution License, which permits unrestricted use, distribution, reproduction and adaptation in any medium and for any purpose provided that it is properly attributed. For attribution, the original author(s), title, publication source (PeerJ) and either DOI or URL of the article must be cited.
License URL: https://creativecommons.org/licenses/by/4.0/

Keywords: Metabolic syndrome, Helicobacter pylori, Insulin resistance, Inflammatory factors

Funding: Joint Medical Research Project of Chongqing Municipal Science and the Technology Bureau and Health Commission 2023QNXM044 Scientific research project of Chongqing Municipal Sports Bureau C202108,C202202,C202206 Chongqing Key Laboratory of Emergency Medicine 2022KFKT03 This work was supported by the Joint Medical Research Project of Chongqing Municipal Science and the Technology Bureau and Health Commission (2023QNXM044), the Scientific research project of Chongqing Municipal Sports Bureau (C202108,C202202,C202206), and the Chongqing Key Laboratory of Emergency Medicine (2022KFKT03). The funders had no role in study design, data collection and analysis, decision to publish, or preparation of the manuscript.

==============================
Background

Globally, metabolic syndrome (MS) and Helicobacter pylori (HP) infection, which have gained an epidemic status, are major challenges to human health, society, and medical professionals. Recent studies have demonstrated that MS is closely related to HP infection. Additionally, HP is an important risk factor for gastric cancer. However, systematic reviews on HP are lacking. This review aimed to summarize and analyze the potential correlation of HP infection with MS and its components, as well as the underlying mechanism, to provide reference and strategies for clinical prevention and treatment.

Methodology

Previous studies examining the correlation between HP and MS since 1990 were retrieved from the PubMed, Web of Science, and Embase databases. The potential correlation between HP infection and MS and its components was comprehensively analyzed. The keywords “Helicobacter pylori,” “HP,” “metabolic syndrome,” “hypertension,” “obesity,” “diabetes,” or “dyslipidemia” were used in all fields. No language restrictions were imposed.

Results

MS was strongly correlated to HP infection. The inflammatory response and inflammatory factors produced during HP infection are important etiological factors for insulin resistance and MS. The co-occurrence of long-term chronic inflammation and immune dysfunction with MS may be the predisposing factor for HP infection. MS components, such as diabetes, hypertension, dyslipidemia, and obesity were also correlated with HP infection in one or both directions.

Conclusions

HP infection and MS may promote the pathogenesis of each other. The contribution of HP infection and MS to gastric cancer cannot be ruled out based on co-occurrence. The MS components diabetes and obesity may be bidirectionally correlated with HP infection.

Introduction

Metabolic syndrome (MS) is a common clinical condition without a global unified definition. However, various associations and guidelines have reached a consensus on its components, namely metabolic aberrations characterized by hyperglycemia, hypertension, dyslipidemia, and obesity. MS affects approximately 25% of the global population (Saklayen, 2018). The clinical outcomes of MS and its components are a major challenge to the global healthcare system, society, and public health. The global number of cases of Helicobacter pylori (HP) infection, which is one of the most common bacterial infections, is as high as approximately 4.4 billion. The incidence of HP infection in developing countries is higher than that in developed countries (Burucoa & Axon, 2017; Hooi et al., 2017). Previous studies have demonstrated that HP infection is an important risk factor for chronic gastritis, gastric ulcer, gastric cancer, and other gastric diseases and that it is closely related to MS (Upala et al., 2016). Based on the disease burden and multiple related complications, MS and HP infection are major health concerns for the global population. The potential correlation between MS and HP infection will have important implications for the prognosis of patients with MS, HP infection, or both MS and HP infection.

Currently, limited studies have examined the correlation of MS and its components with HP infection. Additionally, a comprehensive analysis of the complex causal relationship between MS and HP infection has not been previously performed. This review describes the correlation of MS and its components with HP infection, especially the potential causal relationships and their possible mechanisms. Additionally, potential therapeutic targets have been suggested to enable clinicians and medical researchers to address the co-occurrence of these pathological conditions and consequently prevent the occurrence and development of diseases and related complications and improve the prognosis of MS and HP infection.

Survey methodology

In this study, literature examining the correlation between HP and MS since 1990 was retrieved from the PubMed, Web of Science, and Embase databases. Additionally, the potential correlation between HP infection and MS and its components, as well as the underlying mechanisms, were comprehensively analyzed. The free keywords “Helicobacter pylori,” “HP,” “metabolic syndrome,” “hypertension,” “obesity,” “diabetes,” or “dyslipidemia” were used in all fields. No language restrictions were imposed.

HP and MS

Studies in different countries and regions have demonstrated a positive correlation between HP infection and MS (Azami et al., 2021; Chen et al., 2019b; Gunji et al., 2008; Lim et al., 2019). The prevalence of MS in serum HP-positive patients (27.2%) was significantly higher than that in serum HP-negative patients (21.0%) (P < 0.001) (Lim et al., 2019). HP infection promotes chronic inflammation and immune responses in the stomach and digestive tract in which several inflammatory cytokines and adipokines, such as tumor necrosis factor-α (TNF-α) and leptin are involved (Crabtree, 1996; Ernst et al., 1994; Upala et al., 2016). Compared with those in patients without HP infection, the TNF-α levels are upregulated and the leptin levels are downregulated in patients with HP infection (Kusters, van Vliet & Kuipers, 2006; Roper et al., 2008). The upregulation of TNF-α and the downregulation of leptin can lead to insulin resistance (Matthaei et al., 2000; Shoelson, Herrero & Naaz, 2007). Additionally, HP infection can also induce systemic inflammatory responses, resulting in the upregulation of interleukin (IL)-1, IL-6, IL-8, C-reactive protein (CRP), TNF-α, leptin, and other inflammatory factors and cytokines and the development of insulin resistance (Franceschi et al., 2014; Mansori et al., 2020; Yu et al., 2019). Insulin resistance, which is one of the core pathophysiological mechanisms underlying MS pathogenesis (Cho et al., 2022) (Fig. 1), promotes the release of free fatty acids in adipose tissue, the upregulation of very-low-density lipoprotein, and the downregulation of high-density lipoprotein (HDL). The upregulation of free fatty acids, inflammatory cytokines, adipokines, and mitochondrial dysfunction results in impaired insulin signal transduction, decreased glucose uptake by skeletal muscles, increased hepatic gluconeogenesis, and β cell dysfunction, leading to the development of hyperglycemia. Additionally, insulin resistance promotes the development of hypertension by disrupting nitric oxide-induced vasodilation (Gallagher, Leroith & Karnieli, 2010). Therefore, HP infection may promote the development of MS by inducing insulin resistance.

Figure 1 HP stimulates the digestive tract and systemic circulation to release inflammatory factors and cytokines, leading to insulin resistance and MS.

The effects of MS and insulin resistance on HP infection have not been previously reported. However, we hypothesized that the correlation of MS and insulin resistance with HP infection is bidirectional. In patients with MS exhibiting long-term insulin resistance, the ability of MS to promote HP infection cannot be ruled out. Long-term chronic inflammation and immune dysfunction in patients with MS are potential susceptibility factors for HP infection. Further studies are needed to clarify these correlations.

Recent studies have demonstrated that MS is an important risk factor for tumors, especially for the occurrence and development of gastric cancer and other digestive system tumors (Belladelli, Montorsi & Martini, 2022). HP infection and MS may exert similar effects on gastric cancer. Thus, the synergistic effects of HP infection and MS on gastric cancer cannot be ruled out. Patients with both MS and HP infection should be carefully monitored, especially based on stable metabolic parameters and other controllable factors to reduce the risk of gastric cancer and HP infection.

The effect of HP eradication on MS is currently controversial. Mokhtare et al. (2017) demonstrated that the eradication of HP significantly alleviates the dysregulated fasting blood glucose and HbA1c levels, dyslipidemia, enhanced abdominal circumference, and other important components of MS. Chopeĭ et al. (2014) demonstrated that the eradication of HP can significantly alleviate dysregulated glucolipid metabolism, liver function, and CRP level in patients with MS. Liou et al. (2019) demonstrated that although the eradication of HP did not significantly affect the prevalence of MS, it exerted significant beneficial effects on metabolic parameters, including the alleviation of insulin resistance, the downregulation of triglyceride (TG) and low-density lipoprotein cholesterol (LDL) levels, and the upregulation of high-density lipoprotein cholesterol (HDL) levels. Therefore, several studies have demonstrated that HP eradication alleviates metabolic parameters. However, the effect of HP eradication on the prevalence of MS must be evaluated as it depends on various confounding factors, such as the current diagnostic criteria for MS, the degree of improvement of MS components, and the follow-up time after successful HP eradication. MS standards must be unified, and long-term, prospective, randomized, controlled studies must be designed to further clarify the effect of HP eradication on MS.

Some studies have also reported negative results. A cross-sectional survey revealed no correlation between serum HP antibody positivity and the occurrence of metabolic diseases. This negative result may be related to factors, such as research methodology, system error, and sample size. Thus, further studies are needed to clarify the correlation (Wawro et al., 2019).

HP and diabetes

The main manifestation of hyperglycemia, a component of MS, is diabetes mellitus (DM). Several studies have reported a positive correlation between HP infection and diabetes. HP infection, which affects the incidence of type 1 diabetes, type 2 diabetes, and gestational diabetes, increases the risk of diabetes by approximately 27% (Mansori et al., 2020; Wan et al., 2020; Xia et al., 2020). Meanwhile, the rate of HP infection in patients with diabetes was significantly higher than that in patients without diabetes (Chen et al., 2019a). Additionally, the HbA1c level in patients with HP infection was significantly higher than that in patients without HP infection (Chen et al., 2019a). HP infection was positively correlated with HbA1c upregulation in patients with diabetes (Chen et al., 2019a). Therefore, insulin resistance resulting from HP-induced inflammatory factors is a key etiological factor for diabetes. Additionally, the HP infection-induced dysregulation of cellular and humoral immunity is an etiological factor for diabetes (Borody et al., 2002).

Kim et al. (2022) indicated that HP eradication downregulated the HbA1c levels in patients with diabetes or pre-diabetes. The HbA1c level in the HP-eradicated group was significantly lower than that in the non-eradicated group in the first and fifth years of follow-up. Additionally, HP eradication alleviated hypertension and decreased body mass index (BMI). This may be related to the mitigation of systemic inflammation and insulin resistance. Cornejo-Pareja et al. (2019) demonstrated that HP eradication promotes the secretion of glucagon-like peptide-1 (GLP-1) by modulating the distribution of human intestinal flora. This is one of the potential reasons for the improvement of glucose metabolism after HP eradication. Finally, HP infection is reported to be related to proteinuria in patients with diabetes. HP infection eradication may be an important strategy to protect renal function in patients with diabetes (Shi et al., 2018) although the specific underlying mechanisms are unclear.

Song et al. (2021) demonstrated that the probability of HP eradication failure in patients with DM was two times higher than that in patients without DM. Additionally, the BMI was directly proportional to the HP eradication failure rate. Nam et al. (2019) suggested that poor glycemic control hinders the eradication of HP. Diabetes is an important risk factor for local or systemic infection, and hyperglycemia promotes microbial reproduction. The progression of diabetes is associated with decreased gastrointestinal peristalsis, impaired gastric acid secretion, delayed gastric emptying, and gastroparesis, which increase the risk of microbial colonization and infection in the digestive system (Jeon et al., 2012). Therefore, diabetes can increase the risk of HP infection. The adverse interaction between DM and HP infection cannot be ruled out. DM affects the incidence of HP infection and the efficacy of HP eradication treatment, while HP affects the incidence and metabolic parameters of DM.

Diabetes also increases the risk of gastric cancer development and gastric cancer-related mortality (Tseng, 2021). This suggests that patients with both diabetes and HP infection must be screened for the occurrence and development of gastric cancer.

HP and hypertension

The pathogenesis of hypertension, a component of MS, mainly involves vascular endothelial dysfunction, sympathetic nerve impairment, and the aberrant activation of the renin-angiotensin-aldosterone system (Gupta et al., 2022). Hypertension is also closely related to HP infection (Dore et al., 2022; Fang, Xie & Fan, 2022; Kountouras et al., 2022). Yue et al. (2022) reported that HP infection increased the risk of hypertension by 32%. Wan et al. (2018) demonstrated that HP infection was positively correlated with the risk of hypertension. Additionally, the average arterial pressure of patients with HP infection increased by 0.723 mmHg when compared with that of patients without HP infection. Migneco et al. (2003) revealed that the blood pressure of patients with hypertension after HP infection eradication was significantly lower than that at the baseline. Additionally, the 24-h average systolic blood pressure decreased from 135.5 ± 9.7 to 123.6 ± 11.8 mmHg, while the 24-h average diastolic blood pressure decreased from 85.6 ± 9.3 to 69.3 ± 7.8 mmHg after HP eradication. However, in patients with hypertension in whom HP was not successfully eradicated, the blood pressure at follow-up was not significantly different from that at the baseline (Migneco et al., 2003).

These findings indicated that HP infection is a potential cause of hypertension. Although no specific mechanism for this phenomenon has been reported, some theoretical evidence supports this hypothesis. HP infection-induced inflammatory response and inflammatory factors can damage the vascular endothelium. Dysfunctional vascular endothelium is an etiological factor for hypertension. Additionally, HP infection can affect the immune system and induce autoimmune reactions, which may also cause hypertension (Huang et al., 2021). The colonization of HP has been detected in the arterial wall (Kowalski, 2001). The direct injury to the vessel wall caused by HP infection may impair vascular elasticity and hemodynamics and consequently cause hypertension. HP infection is closely related to dyslipidemia. Long-term dyslipidemia-induced atherosclerosis is also an important factor affecting the occurrence and development of hypertension.

The role of hypertension in HP infection has not been examined. We hypothesize that hypertension and HP infection lack the bidirectional correlation as observed in diabetes. One exception may be hypertension-induced injury of the digestive system, especially the stomach, which increases the susceptibility to HP infection. However, this phenomenon has rarely been observed.

HP and dyslipidemia

Dyslipidemia, a component of MS, is an important risk factor for cardiovascular diseases. Long-term dyslipidemia can significantly increase the risk of acute cardiovascular and cerebrovascular events. Tali et al. (2022) reported that the incidence rates of hypercholesterolemia, hypertriglyceridemia, high LDL, and low HDL in patients with HP infection (73.21%, 68.75%, 77.04%, and 69.72%, respectively) were significantly higher than those in patients without HP infection (26.79%, 31.25%, 22.96%, and 30.28%, respectively). The TG and LDL levels in patients with HP infection were significantly higher than those in patients without HP infection. Similarly, the TC and HDL levels were downregulated in patients with HP infection (Tali et al., 2022). Nigatie et al. (2022) demonstrated that the prevalence of dyslipidemia in patients with HP infection was as high as 71.8%. The median TG, TC, and LDL levels in patients with HP infection (93, 143, and 108 mg/dL, respectively) were significantly higher than those in patients without HP infection (83, 125, and 95 mg/dL, respectively) (Nigatie et al., 2022). Previous studies have demonstrated that HP infection was positively correlated with TC, TG, and LDL levels and negatively correlated with HDL levels (Shimamoto et al., 2020).

Park et al. (2021) reported that the incidence of dyslipidemia in HP-eradicated cases (98.1 per 1,000 person-years) was significantly lower than that in patients with HP infection (130.5 per 1,000 person-years). Additionally, HP eradication decreased the risk of dyslipidemia. Other studies have demonstrated that HP eradication increased HDL levels and decreased TG and LDL levels (Iwai et al., 2019; Mokhtare et al., 2017). Dyslipidemia, especially high LDL and low HDL levels, is closely related to the occurrence and development of atherosclerosis and coronary heart disease. Thus, HP eradication can exert beneficial effects in patients with dyslipidemia.

Pih et al. (2021) demonstrated that LDL upregulation and HDL downregulation were associated with the risk of gastric cancer. These individuals with aberrant blood lipids were not tested for HP infection. If these patients had HP infection, it may be due to the impact of cancer rather than aberrant blood lipids. However, the influence of lipid components on gastric cancer cannot be ruled out. HDL is reported to exhibit anti-inflammatory and antioxidant activities in tumors (Soran, Schofield & Durrington, 2015).

These findings suggest that HP infection may affect the lipid profile through some mechanisms. Although the exact mechanism is not clear, it can be speculated. HP promotes the production of some inflammatory factors (such as TNF-α and IL-8), which can lead to the dysregulation of lipoproteins and lipids and the mobilization of lipids from tissues to blood (Gallin, Kaye & O’Leary, 1969; Park et al., 2021). Additionally, HP infection promotes the release of ghrelin and leptin, affecting the absorption of nutrients and consequently modulating the lipid profile (Shimamoto et al., 2020). Similar to hypertension, the effect of dyslipidemia on HP infection has not been examined. We believe that the possibility of dyslipidemia affecting HP infection is low.

HP and obesity

Obesity is closely related to other components of MS. The etiology and pathogenesis of obesity intersect with those of diabetes, hypertension, and dyslipidemia. Compared with that of the other three components, the risk of obesity is the highest among the general population. Baradaran et al. (2021) reported that the prevalence of obesity is positively correlated with HP infection. The risk of HP infection in patients with obesity is 46% higher than that in patients without obesity. Additionally, patients with HP infection are more prone to develop obesity than non-infected patients (Baradaran et al., 2021). One study demonstrated that HP infection is positively correlated with a high level of BMI and that HP infection is negatively correlated with BMI decline in patients with obesity (Zhang et al., 2020). Obesity is also associated with gastric cancer. The incidence of gastric cancer in patients with persistent obesity increased by 20% (Lim et al., 2022). This phenomenon may be related to the high HP infection rate and insulin resistance. However, the effects of obesity-related inherent factors, such as aberrant adipocyte cytokines, ectopic fat deposition, changes in gut microbiota, chronic inflammation, and oxidative stress on tumors cannot be ruled out (Avgerinos et al., 2019).

These findings indicate the correlation between obesity and HP infection. HP infection promotes the production of inflammatory factors, leading to a chronic systemic inflammatory state and insulin resistance, which can promote the occurrence and development of obesity. The aberrant secretion of serum leptin after HP infection delays the feeling of satiety, increasing energy intake and causing obesity (Baradaran et al., 2021). Additionally, patients with obesity exhibit imbalanced autoimmunity, impaired ability of monocytes to transform into macrophages, and suppressed cytotoxicity of natural killer cells. These factors may promote the occurrence of HP infection (Marti, Marcos & Martinez, 2001; Moulin et al., 2009). Similar to diabetes, the association between obesity and HP infection may be bidirectional (Fig. 2).

Figure 2 Association of HP infection with MS and its components. HP infection may interact with MS, diabetes, and obesity.

Conclusions

HP infection is closely related to MS. Additionally, HP infection may be correlated with the components of MS, such as diabetes and obesity. HP eradication enables the suppression of the occurrence and development of MS and its components, as well as the alleviation of the dysregulated metabolic parameters in vivo. Patients with MS and its components must be screened for HP infection to improve their metabolic status.

Figure support was provided by Figdraw.

Additional Information and Declarations

Competing Interests

Author Contributions

Data Availability

The authors declare that they have no competing interests.

Qinli Xie conceived and designed the experiments, analyzed the data, prepared figures and/or tables, authored or reviewed drafts of the article, and approved the final draft.

Yangjun He conceived and designed the experiments, performed the experiments, analyzed the data, authored or reviewed drafts of the article, and approved the final draft.

Danni Zhou performed the experiments, analyzed the data, prepared figures and/or tables, and approved the final draft.

Yi Jiang conceived and designed the experiments, authored or reviewed drafts of the article, and approved the final draft.

Ying Deng performed the experiments, prepared figures and/or tables, and approved the final draft.

Ruoqing Li conceived and designed the experiments, authored or reviewed drafts of the article, and approved the final draft.

The following information was supplied regarding data availability:

This is a literature review.

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
