# Peer review of "Recent research progress on the correlation between metabolic syndrome and Helicobacter pylori infection"

_PeerJ, doi:10.7717/peerj.15755_

## Round 0.1 · original submission · Major Revisions

Please carefully read the comments and address these questions and concerns.

Reviewer 1 ·

Basic reporting

The manuscript described the relationship and possible interactions between MS and HP infections, and belongs to interdisciplinary research on common diseases, which meets the requirements of the journal. I conducted research in this field before reviewing the manuscript and provided objective comments on the article based on it. The author has fully explained the research background and significance of MS and HP infection in the introduction section, and pointed out that their research can provide a certain degree of support for doctors and researchers.

Experimental design

The author's literature search on MS and HP infections is comprehensive, and the research results have been fully demonstrated and logical. The reference citation of the manuscript is standardized and the source is clear. The manuscript structure is reasonable and logical.

Validity of the findings

The research purpose in the introduction was demonstrated and developed with sufficient evidence in the following text. The research purpose in the introduction had been demonstrated and developed with sufficient evidence in the following text. The research results confirmed the direction of further clarifying the possible causal mechanism between MS and HP infection in the future.

Additional comments

1.The draft should briefly introduce the definition and standards of metabolic syndrome.
2.The correlation analysis and description of MS and HP infection in the manuscript are relatively sufficient, but the results are almost all positive. Is there currently a negative correlation or no relevant research report?
3.The manuscript has already mentioned that both MS and HP infections are important factors in tumor development. Diabetes is one of the components of MS, whether there is diabetes in HP infection on tumor related reports.
4.Hypertension is also an important component of MS, and the view that there is a connection between hypertension and HP infection is interesting. The manuscript has analyzed the possible reasons for their connection, but there is a lack of relevant epidemiological research results on hypertension combined with HP infection. Is there any relevant report currently?
5.The analysis of the relationship between dyslipidemia and cancer mentioned in the manuscript by Pih GY et al. is unclear. Please reorganize the language and describe it in order to understand.
6.The description of obesity and HP infection in the draf mentioned that obesity increases tumor risk. The author suggested that this phenomenon may be related to the high infection rate of HP and insulin resistance. This viewpoint is logically supported, but it is still necessary to list other possible factors that obesity may affect tumors in the following text.

Reviewer 2 ·

Basic reporting

This manuscript mainly discussed the relationship between metabolic syndrome and Helicobacter pylori infection, and the research covers two professional fields: endocrine diseases and digestive diseases, which is in line with the journal's scope of submission.The Introduction adequately introduced the subject and make it clear who the audience are clinical doctors and medical researchers.

Experimental design

The survey methodology consistent with a comprehensive, unbiased coverage of the subject.Sources are adequately cited. The review is organized logically into coherent paragraphs and subsections.

Validity of the findings

There is a well developed and supported argument that meets the goals set out in the Introduction.The conclusion identify future directions.

Additional comments

1.This review mentioned that HP infection can lead to insulin resistance, which is an important core pathogenesis of MS. It is necessary to supplement how insulin resistance affects the occurrence and development of MS.
2.The author summarized the direct relationship between dyslipidemia and HP infection, but lacked possible reasons for these results. If there are currently research reports that can provide relevant evidence, you can try to analyze the reasons.
3.The author analyzed that the relationship among HDL, LDL, and tumors may be due to HP infection. The explanation here cannot be understood, please reorganize the language and logic.
4.Line56-57:The description of the ' global HP infection rate ' should be 'Global infection rate of HP'.
5.The conclusion of the draft reveals that eradicating HP is beneficial for improving the occurrence and development of metabolic syndrome and its components, as well as correcting the disorder of metabolic parameters in the body. Actually, there are other values. For example, it has an enlightening effect on clinical work. When metabolic syndrome and its components are poorly controlled, screening for HP infection should be considered in order to better improve metabolic status.The authors may consider summarizing these appropriate descriptions.

---

## Round 0.2 · accepted · Accept

The authors have addressed the concerns of reviewers. The review paper may be accepted in its current state.